



# How runoff components affect the export of DOC and nitrate: a long-term and high-frequency analysis

Michael P. Schwab[1,2], Julian Klaus[1], Laurent Pfister[1] , Markus Weiler[2]

[1]Catchment and Eco-Hydrology Research Group, Luxembourg Institute of Science and Technology, Belvaux, 4422, Luxemburg
[2]Hydrology, Faculty of Environment and Natural Resources, University of Freiburg, Freiburg, 79098, Germany

*Correspondence to*: Michael P. Schwab (michaelschwab.fr@gmx.de)

**Abstract.** We monitored dissolved organic carbon (DOC) and nitrate concentrations and fluxes in situ with a UV-Vis spectrometer for two years at a high temporal resolution of 15 minutes in the forested Weierbach headwater catchment. The catchment exhibits a characteristic double peak runoff response to incident rainfall during periods with wet initial conditions. When initial conditions are dry, only the first discharge peak occurs. During our observations, both DOC and nitrate concentrations increased during the first discharge peak, while only nitrate concentrations were elevated during the second discharge peak. Relying on additional biweekly end-member data of precipitation, throughfall, soil water and groundwater, we linked the first peak to near surface flowpaths and the second peak to shallow groundwater reactions and subsurface flowpaths. The mass export of DOC and nitrate is largely controlled by the discharge yield. Nevertheless, this relationship is altered by changing flowpaths during different wetness conditions in the catchment. Due to the absence of second discharge peaks during dry conditions, the DOC export is more relevant and the nitrate export is less relevant during dry catchment states. The study highlights the benefits of in-situ, long-term, and high-frequency monitoring for comparing DOC and nitrate export with runoff components that are changing rapidly during events as well as gradually between seasons.





## 1 Introduction

Nitrate and dissolved organic carbon (DOC) are important biogeochemicals in ecosystems as they play a prominent role in the life cycle of organisms and as they are key to a sustainable management of groundwater and surface water quality. The export of DOC and nitrate from catchments depends on numerous different factors, as well as on their mutual interplay.

Especially the question how DOC and nitrate export is linked to different rainfall-runoff transformation processes is of major importance for catchment management and for understanding the relevant flow generating processes.

Inputs of carbon and nitrogen to catchments happen through atmospheric deposition, urban sources like sewage water, agricultural sources like fertilizers, animal excreta and manure spreading, as well as natural sources like decomposing organisms (Causse et al., 2015; Van Gaelen et al., 2014; Mulholland and Hill, 1997). Transport and transformation of DOC

and nitrate in catchments happen in a complex interlinked system including the land-atmosphere interface, soil, regolith, groundwater and streams (Lohse et al., 2009). At the land-atmosphere interface, atmospheric deposition, throughfall and stemflow play a major role (Levia and Frost, 2006). The soil regolith is a crucial place for the decomposition of organic material, as well as for carbon and nitrogen transformation. Hydraulic properties of soils, landuse, topography and the groundwater recharge rate are key to understand the flow paths of water and subsequently DOC and nitrate transport and

hence export from the catchment. DOC and nitrate can be transformed in soils, groundwater bodies, riparian zones and in streams. While the biogeochemical perspective focuses more on transformation processes and reactions like mineralization and immobilization, the hydrologic perspective emphasizes the transport processes (Lohse et al., 2009). Both perspectives interact with each other, as nutrient export can be controlled by both dominant flow processes (transport limited) and by the availability of transportable nutrients (supply limited) (Lohse et al., 2009; Mulholland and Hill, 1997).

Hydroclimatic factors strongly influence DOC and nitrate export. Many studies reported increased export as a result of rainfall events and associated discharge responses (Alvarez-Cobelas et al., 2008, 2010; Causse et al., 2015; Van Gaelen et al., 2014; Rusjan et al., 2008). This so-called flushing hypothesis is based on sufficiently available nutrients that can be transported via near-surface subsurface stormflow to the stream (Dittman et al., 2007; Hood et al., 2006; Hornberger et al., 1994; Raymond and Saiers, 2010; Weiler and McDonnell, 2006). Besides the fast flow and transport pathways, other flow

paths can also control DOC and nitrate export. Water can reach the stream via lateral flow, shallow groundwater flow and deep groundwater flow with a delay of several days, years, decades or centuries. Which flow path is dominant depends largely on catchment properties but also on catchment conditions like soil moisture content and groundwater levels (Lohse et al., 2009). Apart from direct rainfall-runoff effects, DOC and nitrate export also depend on the season and the plant growth cycles. Flow paths can vary over seasons and the availability of DOC and nitrate can change depending on the yearly life

cycle of organisms (Clark et al., 2004; Helliwell et al., 2007).

In the past, understanding of DOC and nitrate export dynamics and subsequent controls by rainfall-runoff transformation processes were often constrained by monitoring technologies at hand. While long-term monitoring protocols were typically based on weekly sampling intervals, high frequency sampling campaigns were limited to a few single events. Kirchner et al.



(2004) advocated high-frequency, field-deployable auto-analyzers as the way forward to change our view on hydrochemical behavior and processes. Since then, several studies (Aubert and Breuer, 2016; Avagyan et al., 2014; Grayson and Holden, 2012; Huebsch et al., 2015; Jeong et al., 2012; Krause et al., 2015; Strohmeier et al., 2013) made use of the technological progress in biogeochemical monitoring and increased the frequency and duration of measurements. High-frequency

monitoring indeed has clear benefits. Catchments generally exhibit a pulsed and highly nonlinear behavior for flow and solute transport. Consequently, monitoring protocols that are too coarse are likely to miss important information during those pulses or so-called hot moments (Krause et al., 2015). Therefore, high-frequency monitoring can improve biogeochemical flux estimations. It can also deliver a more detailed view on catchment functioning and on the interplay between rainfall-runoff processes and the export of biogeochemicals (Aubert and Breuer, 2016; Kirchner et al., 2004; Krause et al., 2015;

Wade et al., 2012). Several high-frequency *in-situ* biogeochemical monitoring techniques are at hand, such as optical UV-VIS spectroscopy, colorimetry, optical fluorescence spectroscopy and electrochemical detection (Blaen et al., 2016).
Advantages of the UV-VIS spectroscopy are that the high frequency monitoring usually comes along with high precision and accuracy. UV-VIS is based on the spectral absorbance of water. Appropriate algorithms transfer the absorbance signal at specific wavelengths to concentrations of various biogeochemical variables (Blaen et al., 2016). *In-situ* UV-VIS

spectrometers are often used for monitoring organic carbon and nitrate. While this method was used for many years in monitoring and regulating sewage water, it has only recently found its way into hydrological processes research. Jeong et al. (2012) compared the response of DOC and particulate organic carbon (POC) in a forested headwater catchment in South Korea under various hydrologic conditions and found that POC was largely exported at high flow, while DOC is mainly exported at low flow. Grayson and Holden (2012) investigated the behavior of DOC in a bog system in northern England –

focusing on the absorbance at different wavelengths to characterize the specific composition of DOC. Another study on high-frequency DOC behavior was done by Strohmeier et al. (2013) in a small forested catchment in Germany. They focused on DOC sources and DOC export under different flow conditions. Avagyan et al. (2014) evaluated the performance and the application of the UV-VIS spectrometry method for DOC monitoring in a Russian mire complex. They compared it to other DOC measurement methods and concentrated on different absorbance wavelengths. Besides high-frequency studies on

DOC, several studies were targeting nitrate as well. Huebsch et al. (2015) evaluated the performance of UV-VIS spectrometry regarding nitrate and Huebsch et al. (2014) investigated the nitrate response of karst springs in Ireland to heavy rainfall events. Instead of investigating how nitrate is related to rainfall-runoff behavior, Aubert and Breuer (2016) studied diel nitrate cycles in streamflow and its driving factors. Despite efforts made in recent years, the full potential for high-frequency monitoring to unveil how fast and slow rainfall-runoff transformation processes are impacting DOC and nitrate

export still needs to be untapped.
In this study, we combine *in-situ* high-frequency measurements of DOC and nitrate simultaneously in the forested Weierbach headwater catchment in Luxembourg, which is characterized by slate bedrock and periglacial layers. The Weierbach catchment shows a dynamic combination of fast and delayed rainfall-runoff reactions, thus high-frequency monitoring has the potential for capturing the entire range of runoff processes, the resulting runoff components and related




DOC and nitrate export. We hypothesize, that the individual concentration signals of DOC and nitrate allow characterizing the relevant flow paths in the catchment. Furthermore, we hypothesize that DOC and nitrate export are controlled by runoff yield on an annual timescale, while on a seasonal timescale the DOC and nitrate export change, depending on the relevance of different flowpaths with changing wetness conditions in the catchment.

**2 Study Site and Methods**

We measured DOC and nitrate concentration with a UV-Vis spectrometer *in-situ* in the Weierbach creek in northwestern Luxembourg (Figure 1) from December 2013 to November 2015. The Weierbach is a 0.45 km$^2$ small headwater catchment with elevations ranging from 450 to 512 m a.s.l. The stream is deeply incised with steep slopes on the East side and gentle slopes on the West side. The catchment is entirely covered with a mixed forest consisting mainly of beech (Fagus sylvatica)

and a smaller part of spruce (Picea abies). The Weierbach catchment is characterized by slate bedrock and overlying Pleistocene Periglacial Slope Deposits (Moragues-Quiroga et al., 2017). The Cambisol soils are generally shallow with a depth of less than one meter. Long term annual precipitation is approximately 950 mm and exhibits little seasonality. On a long term average, rainfall is highest in December and lowest in April. The temperate, semi-oceanic climate is causing a pluvio-oceanic runoff regime with an annual runoff yield of 50 %. Discharge volume is generally larger in winter than in

summer. (Glaser et al., 2016; Klaus et al., 2015; Martínez-Carreras et al., 2016a; Pfister et al., 2017; Schwab et al., 2016).

The DOC and nitrate concentrations in the Weierbach creek were measured *in-situ* with the UV-Vis spectrometer spectro::lyser (s::can Messtechnik GmbH). The spectrometer measures the light absorption spectrum of the stream water between 220 and 720 nm in 2.5 nm resolution with a xenon flash lamp, 256 photo diodes and a two beam instrument. The spectrometer has an optical path length of 35 mm. The measurement interval was 15 min and the measuring window was

automatically cleaned every 3h using pressurized air that was provided by an air compressor.

The spectrometer probe was attached to a metal plate that was placed on the streambed of the Weierbach creek. The probe was oriented horizontally and in stream direction with the measuring window facing towards the riverbed to avoid direct solar radiation. The energy for the spectrometer and the compressor was provided by five 12 V, 50 Ah batteries that were replaced every two weeks. The spectrometer was manually cleaned at the same interval.





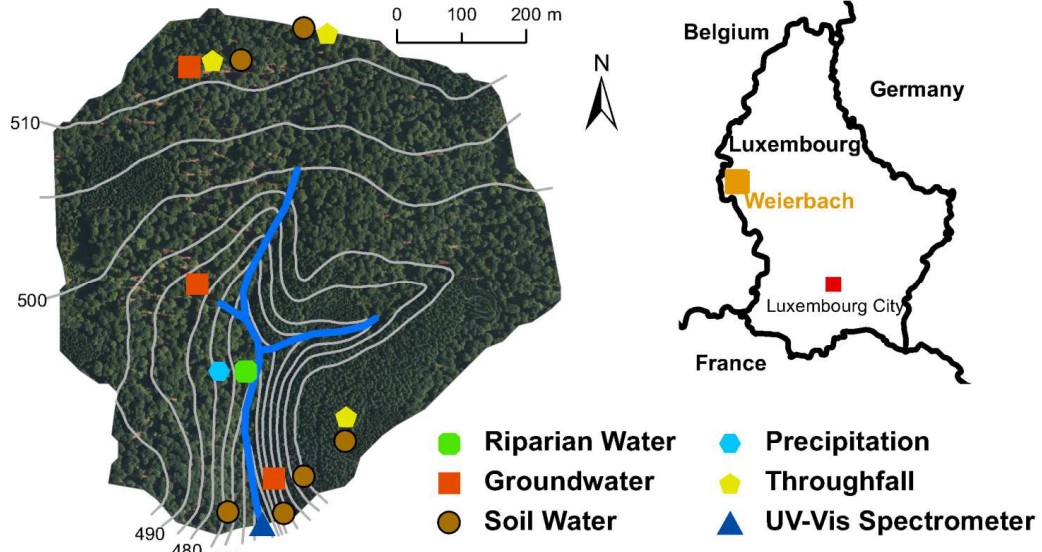

**Figure 1: The Weierbach catchment in Luxembourg with the monitoring locations of the end-members and the UV-Vis spectrometer.**

The spectrometer was operated with a standard global calibration for river water provided by the manufacturer. In addition, a

5 local calibration was derived to adapt the measured parameters to the local conditions and concentrations in the Weierbach creek. Therefore, automatic sampling of several runoff events and weekly to biweekly manual sampling was undertaken. The grab samples were taken to the laboratory as soon as possible, filtered and refrigerated. The samples were analyzed for DOC concentrations with a combustion analyzer (Apollo 9000 - Teledyne Tekmar) and for nitrate concentrations with an ion chromatograph (Thermo Scientific Dionex ICS-5000+ Reagent-Free HPIC). For the calibration of nitrate concentrations,

only manually collected samples were taken into account. The concentrations of the grab samples were compared with the *in-situ* measurements of the spectrometer at the time of the grab samples to perform a local linear regression.

The concentrations of the grab samples covered almost the entire ranges of the observed DOC and nitrate concentrations in the Weierbach creek. The DOC concentrations of the grab samples are linearly correlated with the concentrations measured by the spectrometer ($R^2$ of 0.96). The linear regression for the nitrate concentrations is correlated with an $R^2$ of 0.72 (Figure 2

a and c). No trend for the residuals of both linear regressions is visible over time (Figure 2 b and d).





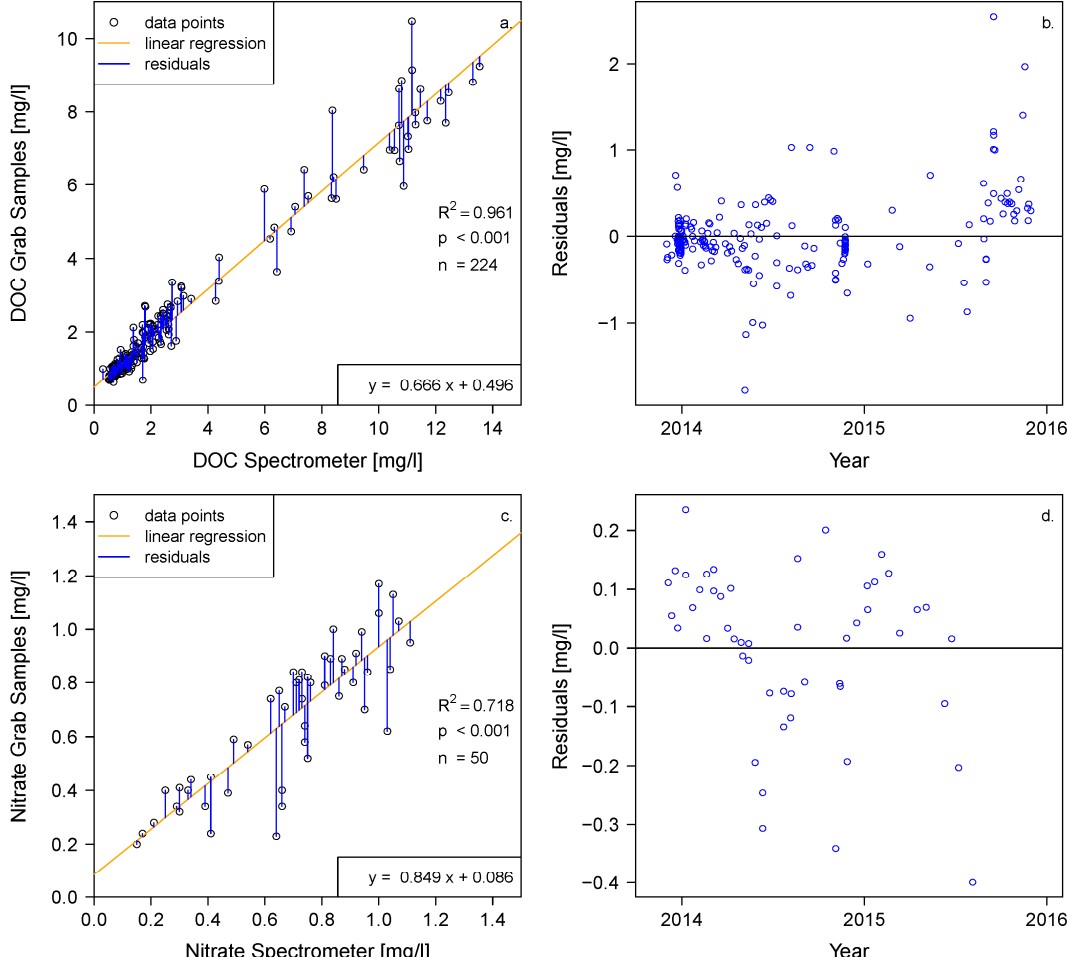

**Figure 2: Calibration of the spectrometer measurements for a) DOC and c) nitrate including the residuals of the linear regression.**

A biweekly time series of end-members and streamflow is available for the Weierbach catchment since 2009 (Martínez-Carreras et al., 2015). Cumulative bulk samples were collected for precipitation at two locations and for throughfall at three locations. Shallow groundwater samples were taken at three wells at a depth of two to three meters that were screened for the lowest 50 cm to 100 cm. Soil water was collected with suction cups in six different locations at depths of 10, 20, 40, 60 and 100 centimeters. The same method was used for riparian water in one location. In addition to the biweekly end-member sampling, a stream water grab sample was taken every two weeks.



Precipitation was recorded with a tipping bucket at the Roodt meteorological station located 3.5 km outside the catchment. Water levels were measured every 5 minutes with a pressure transducer (ISCO 4120 Submerged Probe) and transformed into discharge via a rating curve.

The Weierbach catchment exhibits a characteristic rainfall-runoff response that differs between dry conditions and wet

conditions. During dry conditions, discharge shows a flashy first peak, promptly following the rainfall event. In cases where the rainfall intensity and distribution was not homogenous during the storm event, we observed several fast responding discharge peaks. Those flashy peaks were termed first peak and we called this rainfall-runoff behavior a single peak event. During wet conditions an additional broader second discharge peak appeared after the first peak with a delay of one to several days (double peak event). For analyzing the relationship between rainfall-runoff behavior and the concentration

respectively load of DOC and nitrate in the Weierbach catchment, we divided the discharge time series into three different components: rainfall-runoff events with only a single discharge peak, rainfall-runoff events with a double peak and periods with baseflow. Only events with a minimum rainfall amount of 5 mm were considered and a rainless period with a minimum of 5 hours was necessary to separate two rainfall events. The start of a rainfall-runoff event was assigned to the onset of the rainfall. An event ended 8 hours after the last rainfall or with the onset of a new rainfall event in case of a single peak

rainfall-runoff event. In case of a double peak event, the event ended, when the discharge after the second peak reached the discharge amount at the beginning of the event. In this case, we separated the first peak and second peak periods at the point where the discharge started to increase for building up the second peak. While we identified the first peaks automatically as described above, we selected the second peaks manually based on the hydrograph behavior. The times between rainfall-runoff events were considered as baseflow periods.

While the spectrometer directly measured the DOC and nitrate concentrations, we calculated the DOC and nitrate loads by multiplying the concentrations with the discharge amount and integrating over pre-defined periods. We separated the DOC and nitrate flux during an event into two and four different parts respectively, to compare with the rainfall-runoff behavior: baseflow load during the first peak (B1), event load during the first peak (E1), plus baseflow (B2) and event load (E2) during the second peak in case of a double peak event (Figure 3 a and b). We separated baseflow from event flow by drawing a

horizontal line from the load and discharge value, respectively, at the start of the event (Figure 3 a and b). Negative loads were set to zero.





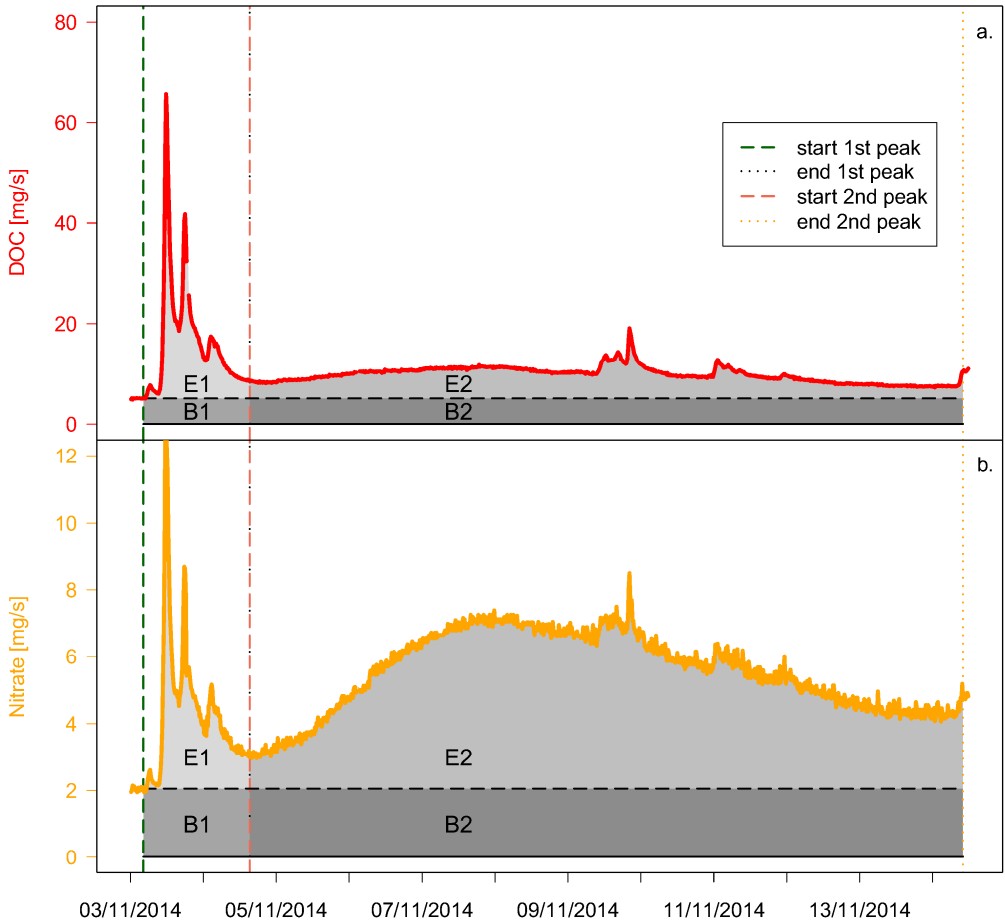

**Figure 3: Separation of DOC and nitrate fluxes into export loads during baseflow (B1) and event flow (E1) for the first peak and baseflow (B2) and eventflow (E2) for the second peak.**

## 3 Results

### 3.1 DOC and nitrate concentrations and their relationship with discharge and runoff components

During the two years of measurement, the discharge generally showed the characteristic behavior of the Weierbach catchment with high flows in winter and early spring and low flows in summer (Figure 4 b). There was a pronounced no





flow period of 2.5 months in summer 2015. This dry period was interrupted by several rainfall events (Figure 4 b and c). In contrast, August 2014 was particularly wet with high discharge. The DOC concentration generally varied between 1 mg l$^{-1}$ and 3 mg l$^{-1}$ and sub-daily peaks could reach almost 10 mg l$^{-1}$ (Figure 4 d). The DOC concentration was lowest in winter and spring and highest in summer and fall. Nitrate concentrations were generally comprised between 0.5 mg l$^{-1}$ and 1 mg l$^{-1}$ and

5   sub-daily peaks reached 2 mg l$^{-1}$ (Figure 4 e). Except for the low flow and no flow period in summer 2015 with flashy nitrate peaks, the nitrate concentration had the highest levels during high flow periods in winter times and in August 2014. The nitrate concentration was decreasing during recession periods, as it can be seen in spring 2014 and 2015 (Figure 4 e).

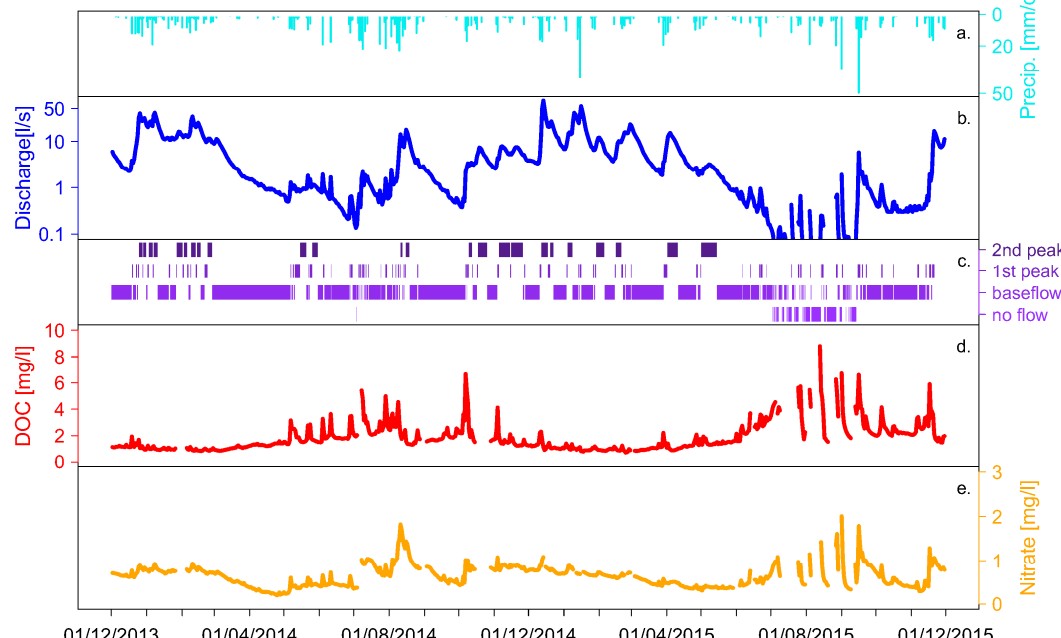

**Figure 4: Two-year time series of daily mean values of a) precipitation, b) discharge, d) DOC concentration and e) nitrate**
10   **concentration plus c) the runoff component periods for no flow, baseflow, first peaks and second peaks.**

Nitrate concentrations were lowest when discharge was low and highest when discharge was high (Figure 5 d), while this pattern was reversed for DOC (Figure 5 c). Separating the concentration time series into the different runoff components, DOC exhibited highest concentrations during first peaks and lowest concentrations during the second peaks (Figure 5 a). Nitrate concentrations were lowest during baseflow periods and similar during first and second peaks (Figure 5 b).



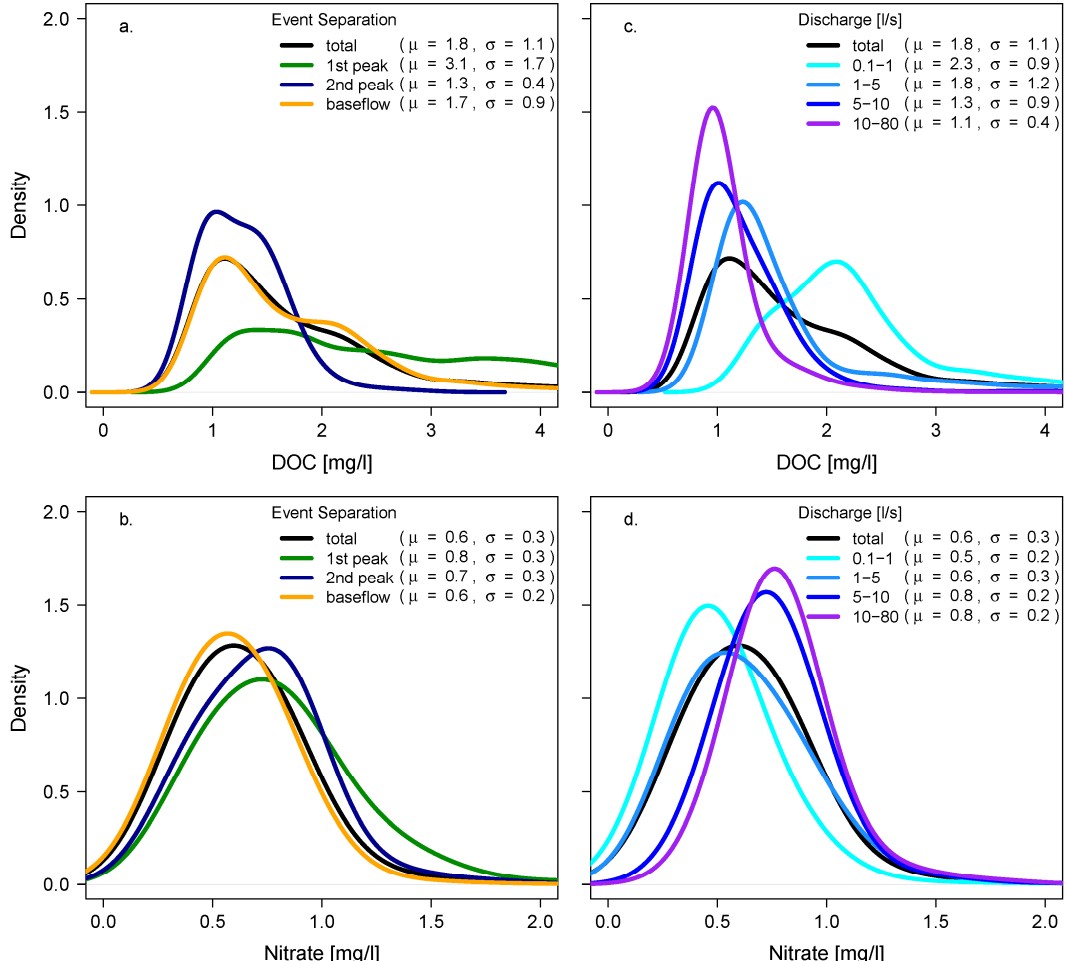

**Figure 5: Probability density plots of DOC and nitrate concentrations separated by runoff periods and discharge volumes. All categories within each subplot are significantly different from each other (significance level of 5%).**

The DOC and nitrate concentrations during rainfall-runoff events were differently linked to discharge and rainfall. Discharge

5    responded to rainfall within 1 or 2 hours with a fast flashy response (see example of two different rainfall-runoff events in Figure 6 a,b,e,f). The DOC and nitrate concentrations followed this fast discharge response during the first peak period simultaneously or with a very short delay (Figure 6 c,d,g,h). In cases where the first discharge peak was not followed by a second delayed peak, the DOC and nitrate concentration decreased similar to the recession of the discharge (Figure 6 f,g,h).





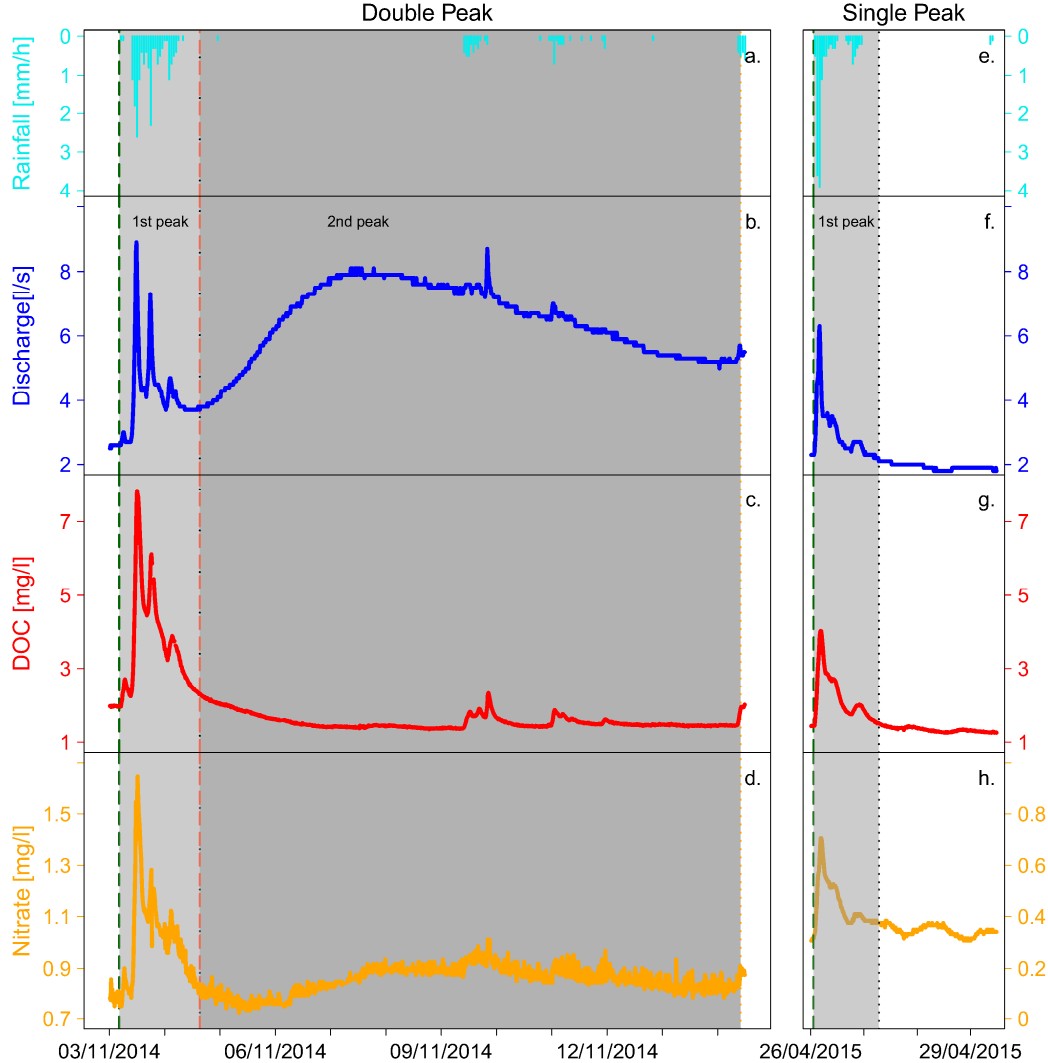

**Figure 6: Characteristic double peak and single peak events and their rainfall, runoff, DOC and nitrate concentration.**

For double peak events, we observed a remarkable pattern concerning DOC and nitrate concentrations. The DOC concentration was not affected by the second peak and fell back to concentration levels that were similar to the initial

5    concentrations at the beginning of the event (Figure 6 d). Nitrate concentration showed a different reaction. The





concentration followed the second discharge peak with a delayed and broad second peak in nitrate concentration (Figure 6 d). The mass transport of nitrate during second peak periods (Figure 3 b) was even more pronounced as loads are a multiplication of discharge and concentrations.

**3.2 Relating streamflow concentration patterns to end-members**

Biweekly sampling of end-members showed apparent differences between DOC and nitrate concentrations. DOC showed high concentrations in throughfall and soil water and low concentrations in groundwater (Figure 7 a). Contrary, nitrate showed the highest concentration in groundwater and only moderately elevated concentrations in soil water and throughfall (Figure 7 c).

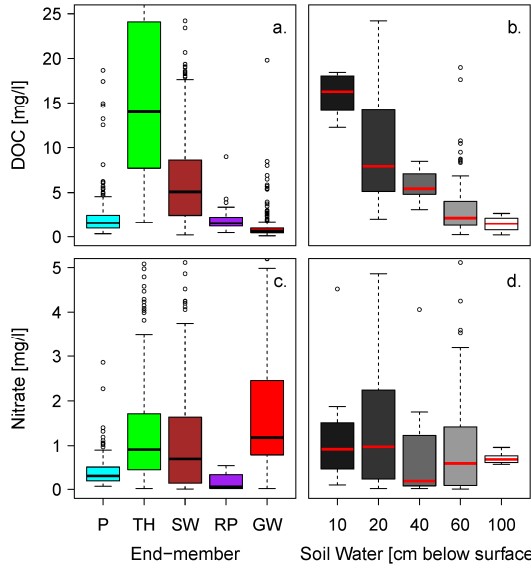

**Figure 7: DOC and nitrate concentrations of the biweekly sampled endmembers and the detailed information for soil water at different depths. P = precipitation, TH = throughfall, SW = soil water, RP = riparian water, GW = groundwater.**

Most of the end-members had DOC and nitrate concentrations that were relatively stable over time. The concentrations were less stable in space, which was especially the case for the soil water depth profiles. The nitrate concentrations were similar at different depths (Figure 7 d), while DOC concentrations showed a decrease over the soil profile from a median concentration

value of 17 mg l$^{-1}$ at 10 cm depth to a median value of 2 mg l$^{-1}$ at 100 cm depth (Figure 7 b).

The dual chemistry plot of DOC versus nitrate concentration (Figure 8) shows a different pattern between measurements during the first peaks, the second peaks and the baseflow periods. During the second peaks, the DOC concentrations stay low, while the nitrate concentrations are increasing. During the first peaks, both the DOC and the nitrate concentrations are




increasing. The baseflow periods show less extreme DOC and nitrate concentrations than during rainfall-runoff events. Looking at the biweekly sampled end-members (Figure 8), the groundwater plots in the corner of low DOC and high nitrate concentration, similarly to the concentrations during the second peaks. The endmembers of soil water and throughfall are plotting in the same direction as the concentrations during the first peak.

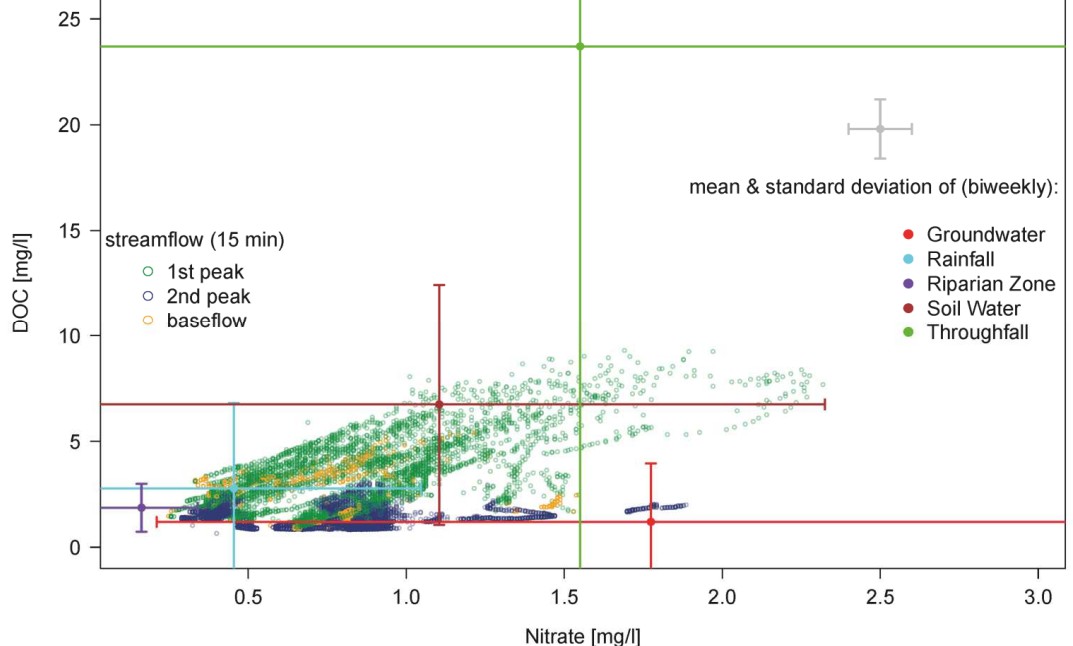

**Figure 8: Dual chemistry plot of DOC versus nitrate concentrations: high frequency spectrometer measurements and biweekly end-member sampling.**

### 3.3 DOC and nitrate fluxes and export

Looking at the dual flux plot of DOC and nitrate (Figure 9), the difference between the first and the second peak is even

10 more pronounced than it is for the concentrations. Nitrate export was more relevant compared to DOC during the second peaks, while DOC export was more important compared to nitrate during the first peak. In addition, hysteresis loops, in particular for the second peaks, can be observed.

Different rainfall-runoff responses (first peaks versus second peaks) were influencing DOC and nitrate export. Yet, this is not the only important factor in explaining the DOC and nitrate export, since also discharge yield controls the export of DOC

15 and nitrate. Figure 10 illustrates the cumulated DOC and nitrate export against the cumulated discharge during the two year period. The total cumulated DOC and nitrate export shows an almost linear relationship with the cumulated discharge.





Although the relevance of discharge volume was clearly visible over the two year time series, some periods showed a different behavior. The periods with lower flows (grey horizontal lines close to each other) deviate from the constant slope between cumulated export and cumulated discharge for both DOC and nitrate. For example, the dry period in the first year from April to July show a steeper slope for cumulated DOC while the slope is flatter for the cumulated nitrate. During those

5    drier periods, second peaks were not occurring or only played a minor role, while the first peaks were more pronounced. Figure 10 illustrates that the export of DOC during the first peaks in these drier periods was more relevant. Obviously, no DOC and nitrate were exported within the second peaks as they did not occur during dry periods. The slope of total cumulated nitrate export during dry periods is flatter, as the second peaks are generally more important for the export of nitrate.

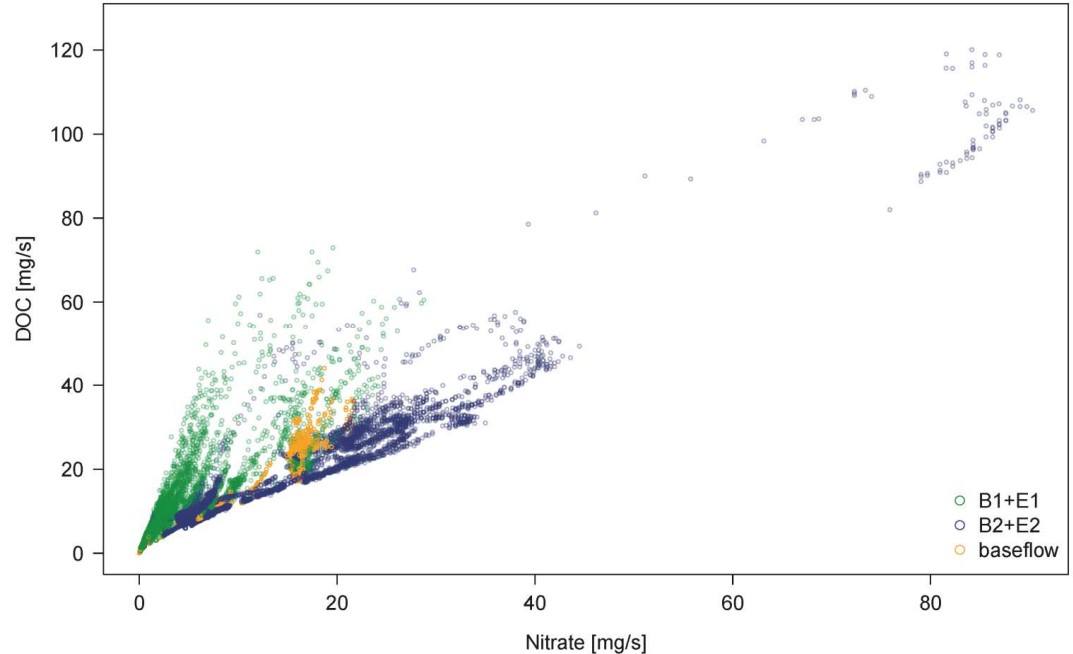

**Figure 9: DOC versus nitrate fluxes separated by first peaks, second peaks and baseflow.**

Concerning the total amount of DOC and nitrate that was exported over two years (Figure 10), the export during the second peaks was similar to the amount that was exported during baseflow conditions (Figure 10 a and b). Contrarily, less DOC and





nitrate were exported during the first peaks (Figure 10 a and b). The relative importance of nitrate export during the first

peaks was smaller compared to the DOC export (Figure 10 a and b).

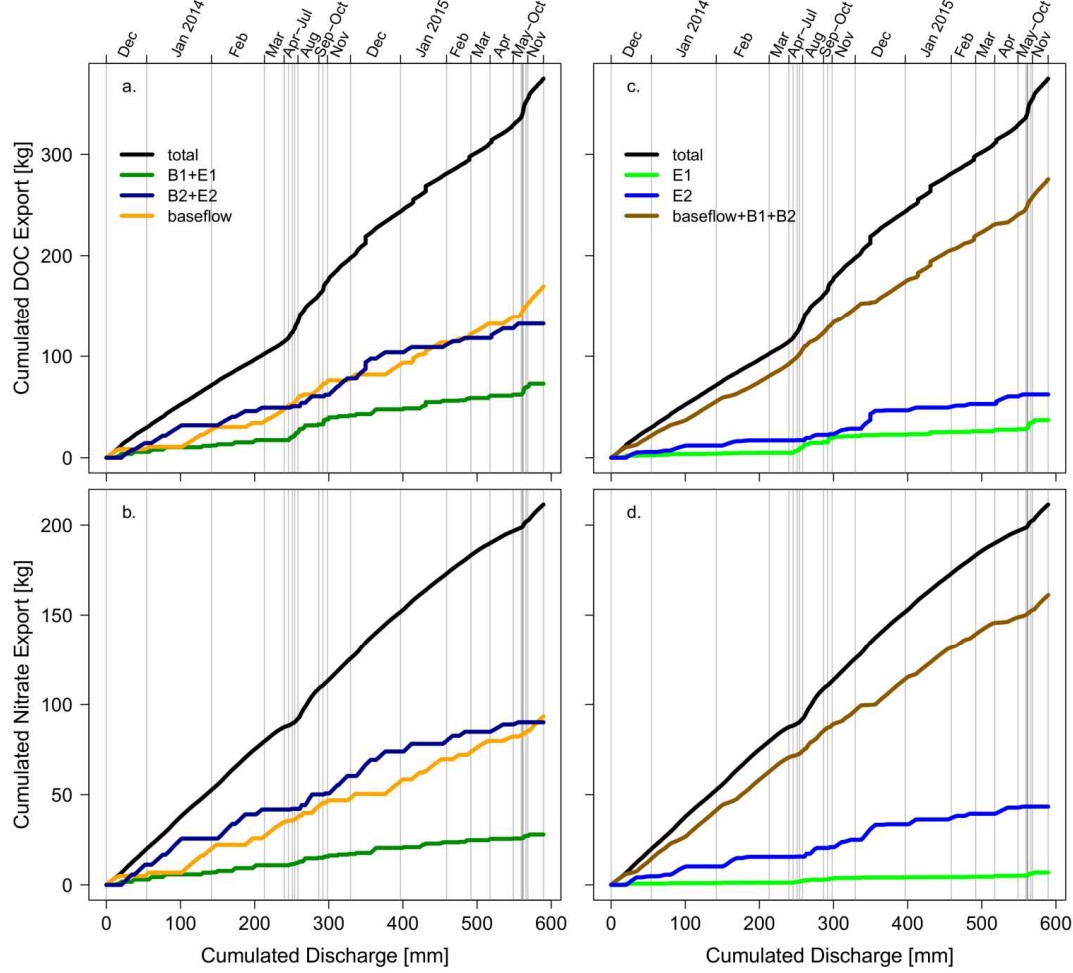

**Figure 10: Cumulated DOC and nitrate export versus cumulated discharge over the two-year period from December 2013 to**
**November 2015. The total cumulated export was split into the cumulated export during the first peaks, second peaks and baseflow**
**periods. Grey vertical lines represent the months; the closer they are, the drier the months.**





## 4 Discussion

We have successfully measured *in-situ* and at high-frequency DOC and nitrate with a UV-Vis spectrometer. We found a good fit between the spectrometer measurements and the grab samples that were analyzed in the laboratory. The forested Weierbach catchment turned out to be well suited for applying the *in-situ* UV-Vis spectrometer method, with nutrient

concentrations, sediment loads and stream temperature being relatively low. When we applied the same method in catchments with higher nutrient concentrations and temperature, algae and biofilms caused a higher maintenance effort of the sensor.

The Weierbach catchment has a characteristic rainfall-runoff behavior with the occurrence of single and double discharge peaks depending on the initial wetness conditions of the catchment. For automatically detecting the first peak of the events,

we predefined selection criteria that were based on the characteristic rainfall-runoff behavior of the Weierbach catchment. However, it was not possible to detect the second peaks automatically. The long duration that characterizes the second peaks turned out to be more challenging. They were occasionally interrupted by a new event, triggering a new second discharge peak that built up on the previous one. Therefore we decided to separate the second peaks visually to reach a more reliable selection of first and second peaks.

We used a simple approach of drawing a horizontal line from the load value at the start of the event for separating baseflow DOC and nitrate flux (B1 and B2) from event flux (E1 and E2) (Figure 3). One could argue that a more elaborated approach is needed to better separate event flux from pre-event flux. However, with the data available, a more elaborated approach would not automatically be more reliable. Therefore we argue that our method is not oversimplified and additionally, it is easily reproducible.

The DOC and nitrate concentrations of the end-members and the concentration dynamics in the streamflow provided a good indication on the different flow paths that were creating the first and the second discharge peak. Based on the elevated soil water concentrations of nitrate and especially of DOC we concluded that the fast first peak was likely created by fast overland flow or near surface flowpaths. This conclusion supports the flushing hypothesis (Weiler and McDonnell, 2006) that was described in several other studies (Dittman et al., 2007; Hood et al., 2006; Hornberger et al., 1994; Raymond and

Saiers, 2010; Weiler and McDonnell, 2006). A complementary explanation for the first peak could be a direct input of throughfall into the stream. For the second peak, nitrate is a good indicator for understanding the underlying flowpaths. During the second peak only nitrate concentrations and not DOC concentrations are increasing. Additionally, the nitrate concentration in the shallow groundwater is elevated. Therefore, we conclude that the second peak was generated by a reaction of the shallow groundwater table of the valley aquifer that likely extends into the hillslopes. An observed rise of the

shallow groundwater table during the second peaks supports this conclusion. This is in agreement with observations and simulations by Glaser et al. (2016). They modeled the hillslope-riparian-stream continuum in the Weierbach catchment with an integrated surface subsurface model (HydroGeoSphere). Furthermore, they explained the second discharge peaks with "subsurface flow through the highly conductive saprolite layers". The subsurface flow is supposedly initiated by either





infiltrating rainfall that causes the groundwater table to rise into the saprolite or by a perched groundwater table. The perched groundwater table may occur at the boundary between the saprolite and the underlying fractured slate that has a lower hydraulic conductivity (Glaser et al., 2016).

We have strong evidence supporting our hypothesis that discharge largely controls the export of DOC and nitrate. The
relationship between cumulated discharge and cumulated DOC, respectively nitrate, is almost linear (Figure 10). Nevertheless, during the dry periods we had to consider the changes in runoff components (no second peak) to understand the export of DOC and nitrate. Due to the absence of the second peaks, a decreased relevance of nitrate export during dry conditions was observed. We observed the opposite for DOC, since first peaks played a larger role during the dry conditions and DOC concentrations were only elevated during first peaks and not during second peaks. These results, together with the
process understanding of the runoff generation during the different runoff components, strongly support our hypothesis that varying runoff components and flow paths can explain differences in DOC and nitrate export depending on the initial wetness conditions. Previous studies in the Weierbach catchment clearly showed that the second discharge peaks only emerged during wet conditions when the soil water and groundwater storages are not depleted  (Glaser et al., 2016; Martínez-Carreras et al., 2016b; Wrede et al., 2015). Which landscape unit controls the second peaks is still debated;
Martinez-Carreras et al. (2016b) highlighted to role of the plateaus in the catchment, while Glaser et al. (2016) stressed the importance of the hillslopes. Both studies have in common that the groundwater storage needs to be connected to the stream before generating the second peaks. The combination between the increased contribution of groundwater during second peaks and the elevated nitrate concentrations in the groundwater may explain the elevated nitrate concentrations in the stream during second peaks. Various studies in the Weierbach catchment explained the first discharge peaks by different
flowpaths and runoff responses. Based on their simulations, Glaser et al. (2016) suspected that saturation excess overland flow generates the first peaks. Klaus et al. (2015) and Martinez-Carreras et al. (2015) identified saturation excess flow generated in the riparian zones as the process creating the first peaks. However, modelling work by Glaser et al. (2016) could not confirm the role of the riparian zone as the catchment unit generating the first peak. Further studies associate the first peak with new, namely event water (Martínez-Carreras et al., 2016b; Wrede et al., 2015). An irrigation experiment in the
vicinity of the Weierbach catchment identified the importance of preferential flow for generating fast runoff responses (Jackisch et al., 2016). All these studies concur to the first hydrograph peaks being linked to flowpaths in the (top)soils. The occurrence of these flowpaths is relatively independent from the wetness conditions of the catchment and we have identified the highest DOC concentrations and somewhat elevated nitrate concentrations in the (top)soils. This may explain the increase in DOC and nitrate concentrations during the first peaks for dry and wet conditions.
The fact that we combined long-term with high-frequency measurements was essential to derive a detailed understanding of the DOC and nitrate export processes and pathways. By only focusing on a few single events in high-frequency we would have missed the relevant seasonal differences. By only focusing on a few single events at high-frequency we would have missed important seasonal differences. Ultimately, had we only monitored at low temporal resolution, we would have missed the hot moments (Krause et al., 2015) of the first peaks.



## 5 Conclusion

*In-situ*, long term, high-frequency UV-Vis spectrometry measurements of DOC and nitrate enabled us to analyze the relationship between the rainfall-runoff behavior and the DOC and nitrate concentration in the streamflow, as well as the export of DOC and nitrate from the Weierbach catchment. The first discharge peak of a rainfall-runoff event was linked to fast near surface runoff and showed both DOC and nitrate concentration peaks in the stream. During wet initial catchment conditions, we observed a delayed second discharge peak with elevated nitrate concentrations, which we linked to subsurface flow and an increase of the shallow groundwater table. The export of DOC and nitrate was largely dependent on the discharge volume. Nevertheless, the absence of a second discharge peak during dry conditions modified this relationship. The export of DOC was more and the export of nitrate was less relevant during dry conditions.

Our study was based on instream DOC and nitrate measurements every 15 minutes over two years. In contrast, the end-member concentrations were only measured biweekly. In general, it would be interesting to measure the end-member concentrations at a much higher temporal resolution during several rainfall events to detect possible high-frequency variations of end-member concentrations. A way forward could be to switch one or several mobile UV-Vis spectrometers between the different end-member measurement locations.

Additionally, it could be interesting to investigate the behavior of rainfall-runoff reactions and the DOC and nitrate concentration and export in more human influenced catchments. Therefore, high-frequency measurements in catchments with a large proportion of agricultural fields or settlements would be of great interest.

### Acknowledgements

We acknowledge the FNR (Fonds National de la Recherche Luxembourg) for having funded this research through the AFR PhD grant (Grant 6931545). Additional funding was provided through the FNR-DFG CAOS-2 project (INTER/DFG/14/02) and the DFG funded CAOS project (FOR 1598). We also thank Jean François Iffly, François Barnich and Jérôme Juilleret for their support during field activities and in the laboratory. A special thanks goes to Christophe Hissler, who provided us with the biweekly end-member dataset that was acquired during the project FNR/CORE/SOWAT (C10/SR/799842).

### Competing interests

The authors declare that they have no conflict of interest.

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
