# Peer review of "How runoff components affect the export of DOC and nitrate: a long-term and high-frequency analysis"

_Hydrology and Earth System Sciences, 2017_

## Referee Comment (RC1) · Anonymous Referee #1 · 30 Aug 2017

This manuscript reports the use of high frequency measurement of DOC and nitrate using UV VIS spectrophotometer in a forested headwater catchment, Weierbach. The scope of the manuscript is relevant to HESS and presents on the use of relatively new tools –high frequency, automated sensor data to understand the dynamics of DOC and nitrate in the catchment. Rigorous evaluation of these emerging technologies is a pressing need. The aim of this manuscript was to demonstrate that these high frequency automated sensor data add value to understanding event and seasonal dynamics of DOC and nitrate in this catchment. Indeed, the manuscript reaches a confirmatory conclusion that the sensors add value in detecting event peaks and seasonal trends. However, at the end of the manuscript, it is not clear to the reader how transferable these methods are to other catchments; the methods and limitations of the sensors remain poorly articulated in this manuscript. In particular, a number of concerns are raised concerning detail of the methods and assumptions. These concerns include 1) details on methods for filtering samples and calibration of sensor seemed inadequate to reproduce, 2) the adequacy of use of linear regression for calibrating sensor data to manual grab samples and well as 3) use of non-conservative tracers as end members to attribute to sourcing. Most studies in the literature use concentrations obtained from grab samples as the known concentration values (X axis) and sensor data as the response variable (Y axis). Uncertainty in both the x and y as well as self-correlations (see Worrall et al. 2015 should be given consideration and other models (possibly orthogonal regression or other models (see Vaughan et al. 2017)) should be adopted to address uncertainty in x and y values. Finally, this manuscript is not as fluent as it could be, and it could be more concise. There were numerous grammatical and tense agreement issues as well as awkward statements that made accessing the results and discussion a bit challenging. I believe that there could be better use of figures and some figures could be combined and/or possibly put into supplemental materials. Standard hydrograph separation techniques should be used or better referenced. Adequate referencing of description of prior research in the Weierbach is needed in the introduction to understand the research gap but also the rationale behind single and double peak separation. Some of these studies are cited in the discussion but this discussion comes too late. Finally, putting these data into context of other studies is desperately needed to improve the manuscript.

Specific comments

Page 4, line 1-5: Hypotheses are a bit awkward; what is meant by individual concentration signals?

Page 5, line 4: Need more details on global calibration provided by manufacturer.

Page 5, line 7: filter through what kind of filter? Need more details here. And were the

samples run for NPOC or TC-TIC? Then you are sparging the samples?

Page 5, line 10: if only using nitrate samples, how did you cover the full range of nitrate concentrations at high discharges?

Page 5, lines 10-15: Most studies found in the literature (e.g. Vaughan et al. 2017) use concentrations obtained from grab samples as the known concentration values (X axis) and sensor data as the response variable (Y axis). This is the reverse of what has been done in Figure 2. Uncertainty in both the x and y as well as self-correlations (see Worrall et al. 2015 should be given consideration and other models (possibly orthogonal regression or other models (see Vaughan et al. 2017)) should be adopted to address uncertainty in x and y values. Were the residuals of the line regression examined to determine if they are normally distributed? They appear at bit heteroscedastic, especially for DOC. Finally, this section is awkward and could use a rewrite. Here the manuscript starts to switch back and forth. For example, it should read DOC concentrations of grab samples "were" linearly correlated to stay consistent with past tense used throughout paragraph.

Page7, line 3: More specifics on rating curve are needed. Are these published rating curves or derived from other publications. If so, please reference or provide more details.

Page 7, line 4-19: This section needs more referencing to prior research in the catchment as well as referencing for hydrograph separation. If these rainfall runoff characteristics are not reported elsewhere (but it seems like they are based on discussion), these are results and should go into the results section. Also, referencing to standard hydrograph separation or using accepted hydrograph separation would add utility to this paper.

Page 7, line 24: Figure 3 could be perhaps moved to supplemental if adequate referencing is use.

Page 12, line 4: here the mansuscript using DOC and nitrate endmembers and infers sources without providing additional conservative tracers to determine sources. Did you collect or analyze samples for other conservative tracers for both hydrograph separation and end member mixing analysis. Using a conservative tracer is a central assumption of these analyses (see Barthold et al. (2011)) such that the validity of interpretation of sourcing in this manuscript is questions without additional evidence supporting this conclusion.

Page 12, line 16. Suggest change paragraph to past tense.

Page 13, line 10, relevant? Suggest a different word choice, suggest substantial

Page 16, line 5: this line refers to other studies and limitation of the sensors. Can you reference to these studies? Perhaps this would be a better method paper if these data were also reported to show the limitation of the sensors?

Page 16, line 8-19: This seems slightly redundant to methods. Perhaps include this section or shortened version thereof in methods. Can you reference other studies that have done this?

Page 16, line 20-34. Again, tenses are switched in the paragraph. I'd suggest keeping in past tense.

Page 17, line 4, suggest delete "have"

Page 17, line 5: awkward sentence—suggest revise

Page 17, line 12 and starting line 20. Here you refer to other studies that have been conducted in the catchment identifying the single and double peaks. This is a strong rationale for your study, approach, and analysis. I think some of this needs to come up to the introduction or methods.

Page 17, line 32: Suggest delete second sentence starting "By only…" repeat of same sentence.

Figures: Figure 8 and 9 could be combined into one figure with different panels.

Technical corrections: Page 4, Line 8: spell out masl the first time

Page 4, line 9 and 10: italicize Fagus sulvatica and Picea abies

Page 4, Line 10: delete period (.) before (Glaser et al., 2016)

Page 4, line 13: Suggest change "The" to "A" and change "is causing: to "results in"

Page 6, line 3, suggest delete "for" before throughfall

Page 6, line 7, suggest add "(Figure 1)" after location

Page 9, line 6: delete "the" before nitrate and change to "had" to "were" and delete "levels".

Page 9, line 7: Suggest sodify sentence to read "Nitrate concentrations decreased during recession periods, as observed in spring 2014 and 2015. . .

Page 11, line 5: suggest change "reaction" to "response" because you can not infer reaction.

Page 16, line 2: delete "have"

References referred to within comments:

Vaughan, M. C. H., et al. (2017), High-frequency dissolved organic carbon and nitrate measurements reveal differences in storm hysteresis and loading in relation to land cover and seasonality, Water Resour. Res., 53, 5345–5363, doi:10.1002/2017WR020491.

Worrall, F., Burt, T. P. & Howden, N. J. K. The problem of self-correlation in fluvial flux data – The case of nitrate flux from UK rivers. Journal of Hydrology 530, 328–335 (2015).

Barthold, F. K., C. Tyralla, K. Schneider, K. B. Vaché, H.-G. Frede, and L. Breuer (2011),

How many tracers do we need for end member mixing analysis (EMMA)? A sensitivity analysis, Water Resour. Res., 47, W08519, doi:10.1029/2011WR010604.

---

## Referee Comment (RC2) · Anonymous Referee #2 · 5 Sep 2017

Manuscript presents results of intensive, 2-year high-frequency monitoring campaign of streamwater DOC and nitrate in a forested catchment in Luxembourg by using UV-Vis spectrometer, which is relatively new and emerging technology in field monitoring systems. Credible assessments of UV-Vis spectrometer advantages and limitations in different hydrological conditions is undoubtedly of high value. By obtaining high-frequency data, authors aimed to identify relevant flow paths in the catchment which regulate individual DOC and nitrate concentration signs.

I find the manuscript in line with aims and scope of HESS. Generally, the paper is well written, however there here are some grammatical issues and weird statements that

made it somewhat difficult to follow the link between the results and the discussion section. The discussion section needs to be improved, related especially to other studies at the same study site which also aimed to identify preferential flow paths during different hydrological conditions. Details are provided bellow.

General comments:

The introduction section provides a good overview of the governing biogeochemical and hydrological processes regulating DOC and nitrate exports.

While mentioning the benefits of high frequency monitoring techniques it would be worth noting that the high frequency water quality measurement are valuable especially in small catchments where hydrological mechanisms usually respond to rainfall inputs very quickly and there is usually a strong interconnection between soil biogeochemical conditions (usually the main controlling factors of DOC and nitrate mobilization or retention) and hydrological processes.

The comparison of grab water sample concentration and in-situ concentrations of DOC and nitrate are very informative. I would suggest the authors to point out in conclusions (related to their experiences) how they suggest to combine UV-Vis measurements with grab water samples. I would also suggest to show 1:1 line in fig. 2a and 2c which would illustrate the agreement between the two datasets. Namely, in the case of nitrate, the regression line is close to 1:1 line whereas in the case of DOC there seems to be quite a discrepancy especially for high DOC concentrations (regardless of the fact that linear correlation is good) which indicates that one should be careful when using UV-Vis concentrations without additional grab sampling control.

What is the proportion (e.g. in %) of the total annual DOC and nitrate flux that is exported by the baseflow and by events as defined in the study? Overall, I agree with the authors opinion that the proposed methodology for separating the baseflow DOC and nitrate from events fluxes is simple and could be used elsewhere. However, the method is in principle based on graphical baseflow separation techniques and is not

something new.

In the Discussion section, the authors refer to other studies at the same experimental catchment. But there is relatively poor discussion of the results in relation to process understanding. I miss more tangible discussion on how the results of the DOC and nitrate fit into other studies mentioned in the discussion that were done at the same study site. Do they agree well or do the show that some of the explanations proposed in other studies are not in line with the results shown here.

Another thing that in my opinion strongly influences preferential flow paths (such as flow paths near the surface or in top soils) is the influence of antecedent wetness, rainfall abundancy and intensity in relation to soli infiltration capacity. Was anything done in this direction? Have authors of this or some other studies in the experimental catchment observed some "boundary conditions" which could be related to the solute concentrations behavior in wet periods and so formation of so called "second peaks"?

Specific comments:

Page 4, lines 16-18: What are the technical characteristics of the UV-Vis spectrometer in terms of the DOC and nitrate concentrations (min, max concentration, detection limits, accuracy, etc.).

Page 7, line 12: The rainfall amount of 5 mm seems rather small in order to be considered as a rainfall events. Any additional comment on rainfall losses and rainfall interception, average monthly evapotranspiration from the forested catchment?

Page 7, lines 9-12: The sentence is unclear and need to be rewritten.

Page 7, line 13: I suggest changing: . . .with a minimum 5 h time gap. . .

Figure 5 caption: Does Fig. 5 really show discharge volumes, units are in m3/s? Page 9, line 14: I suggest changing the statement to: . . .similarly increased during first and second peaks.

Page 11, line 5: Authors mention increase of nitrate concentration during second peak. Looking at Fig. 6d, this increase is very small (from approx. 0.8 mg/l (pre-event concentration) to 0.9-1.0 mg/l. I wonder how can this "slight" increase in nitrate concentration be explained in view of UV-Vis spectrometer accuracy?

Page 14, line 2: Is the comment on the results related to Figs. 10 c and d?

Page 16, lines 1-2: I believe the discussion on the goodness-of-fit between laboratory DOC concentrations ad in-situ UV-Vis concentrations should be further discussed accosting to my comment provided above.

Page 16, lines 5-7: Please add some references (if available) while mentioning potential problems with the use of measuring equipment in environmental settings different than the presented study site.

Page 16, line 2: Was the fit between UV-Vis and lab measurement really good (seem my previous comment regarding DOC measurements)?

Page 16, line 22-23: Are there any field evidences that preferential overland or near surface flow paths really occur at the studied catchment?

Page 16, line 24: The flushing hypothesis was not originally proposed by Weiler and McDonnell (2006), one of the first that proposed the flushing hypothesis were Hornberger et al. (1994). Therefore I suggest changing the order of the listed references.

Page 17, line 9-12: I believe that vice-versa is also true. So the export behavior of DOC or nitrate (or maybe some other dissolved substances) can be very helpful for explaining various runoff components.

Page 17, Line 34: What is meant by "hot moments"?

Page 18, lines 15 – 18: Last paragraph of the Conclusion section seems rather general and is in my view not in line with the main theme of the study.

---

## Referee Comment (RC3) · Anonymous Referee #3 · 1 Nov 2017

Overview: Recent technological developments have enabled watershed researchers to monitor important biogeochemicals at high frequency for long periods. Sensors that measure both carbon and nitrate are uniquely suited to study the coupling of nitrogen and carbon cycles across varying temporal scales. The authors present a high-frequency, multi-year data set of nitrate and dissolved organic concentration over a range of hydrological and climatological conditions in a forested watershed. The authors leverage this high density dataset to examine rain-fall runoff responses of nitrate and DOC over varying climatological conditions. The authors suggest that antecedent moisture conditions, and as a result, groundwater levels, drives the relative fluxes of nitrate and DOC form the basin.

General comments: This study was well conceived and the results are clearly presented. However, several elements of the manuscript need attention before this study warrants publication. 1. Further, claims that seasonal differences in nitrate/doc fluxes were observed, are not clearly supported by the data presented in this manuscript. Rather, initial dryness is the direct driver. Consider including data that clearly links the occurrence of preceding dry conditions to season over a longer period of time (longer than two years) to support the seasonal link to hydrologic conditions over the period of this study. 2. The authors fail to put the implications of the study- that "dry"/"reduced wetness" antecedent conditions results in larger fluxes in nitrate to the stream-into a larger context. 3. Consider removing "long-term", as data collected for less than three years hardly justifies the use of this term. 4. The authors should consider using an outside editing service, given the occurrence of several awkward statements throughout the text (e.g. Pg 2, Line 5; Pg.3 lines 28-30; Pg 12, lines 10-13).

Specific comments: Abstract: Pg 1 Line 11: Define "dry", as this definition is critical in the interpretation of the results as well as applying the findings to other locales and placing the implications of this study into a larger context. Introduction: Well cited, however, consider adding Pellerin et al., given the similarity in use of continuous DOC and nitrate sensors to document varying biogechemical yield over different hydrological conditions in a forested watershed. Methods: I was puzzled as to how the probability density plots were developed, and the source of the data. Please explain in detail which if not all storms were considered and how these distribution plots were generated/modeled. The deployment techniques should be more clearly documented to ease duplication of the study. Was the approach modeled after the used in another study? If so please cite. For example, one important aspect is how the sensor cleaned (Birgand et al., 2016, Etheridge et al., 2013)? Pg 5, line 6-7. What was the typical (or max) holding time before analysis of discrete samples?

Results: Consider moving Figure 1 to supplemental material. Characterize the model fits and explain or speculate when/why the outliers tended to occur, especially high

residuals for DOC in late 2015. Placing discrete sample data on the time series Figure 4, would help interpret the limitation of this measurement strategy, i.e. non-linearity due to fouling, light blockage, high turbidity, etc. Please explain the DOC/nitrate data gaps in the summers of 2014/2015.Are these gaps a result of sensor limitations, or were the data otherwise removed? Also, what is the presumed influence of these missing data on cumulative DOC/NO3 export presented in Figure 10. Figures 3, panel b and Figure 4, panel e and Figure 6, panels d, h, nitrate trace is almost illegible given the color selection. Consider a darker color for the trace and y-axis font. Figure 4 Justify the presentation of was daily mean values rather than another descriptive statistic (mean, max, etc). Pg 9, line3: If the data is sampled to daily mean, how are sub-daily peaks resolved? Pg 11, line 3: replace remarkable to "notable" or equivalent. Pg.11, Line 3-5/Figure 6. Do the authors speculate on the mechanism for a steady decline/recession in DOC, despite the rise in discharge during second peak? Pg 12, lines 10-13: be specific about which end members Consider replacing figure 8, which is unclear and messy and rather plot a few select storms that illustrate hysteresis loops. Pg 12, line 18. Change "increasing" to "variable" Pg 13 Lines 10-13.More detail needed here, e.g. what direction were hysteresis loops, for example? Discussion: Pg 16, lines 5-7.Citation suggested here instead of personal experience not included in this study. Pg 16, line 29-30.Where is the evidence of a rise on groundwater table to support this claim? Pg 17, line 26.Consider changing "concur to the" to "suggests that"

―――――――――――――――――――

---

## Author Comment (AC2) · 20 Dec 2017

Manuscript presents results of intensive, 2-year high-frequency monitoring campaign of streamwater DOC and nitrate in a forested catchment in Luxembourg by using UVVis spectrometer, which is relatively new and emerging technology in field monitoring systems. Credible assessments of UV-Vis spectrometer advantages and limitations in different hydrological conditions is undoubtedly of high value. By obtaining highfrequency data, authors aimed to identify relevant flow paths in the catchment which regulate individual DOC and nitrate concentration signs.

I find the manuscript in line with aims and scope of HESS. Generally, the paper is well written, however there here are some grammatical issues and weird statements that made it somewhat difficult to follow the link between the results and the discussion section. The discussion section needs to be improved, related especially to other studies at the same study site which also aimed to identify preferential flow paths during different hydrological conditions. Details are provided bellow.

General comments:

The introduction section provides a good overview of the governing biogeochemical and hydrological processes regulating DOC and nitrate exports.

While mentioning the benefits of high frequency monitoring techniques it would be worth noting that the high frequency water quality measurement are valuable especially in small catchments where hydrological mechanisms usually respond to rainfall inputs very quickly and there is usually a strong interconnection between soil biogeochemical conditions (usually the main controlling factors of DOC and nitrate mobilization or retention) and hydrological processes.

We appreciate the constructive comments made by the reviewer. Referring to the value for small catchments is a good suggestion that we will include in the manuscript.

The comparison of grab water sample concentration and in-situ concentrations of DOC and nitrate are very informative. I would suggest the authors to point out in conclusions (related to their experiences) how they suggest to combine UV-Vis measurements withgrab water samples.

We can include that in the revised manuscript. It is of importance to take grab samples at different flow stages and during first and second peaks and during base flow conditions.

I would also suggest to show 1:1 line in fig. 2a and 2c which would illustrate the agreement between the two datasets. Namely, in the case of nitrate, the regression line is close to 1:1 line whereas in the

case of DOC there seems to be quite a discrepancy especially for high DOC concentrations (regardless of the fact that linear correlation is good) which indicates that one should be careful when using UV-Vis concentrations without additional grab sampling control.

Showing a 1:1 line might be misleading for the reader, as we did not expect to have a 1:1 agreement. An offset and a slope different from 1 has to be expected when comparing the in-situ spectrometer measurements with grab samples that are analyzed in the lab (the manufacturer of the spectrometer also points this out in the spectrometer manual). Stream water has a different background matrix compared to calibration standards that are used in the lab. Therefore it is essential, to take grab samples directly in the stream to compare them with the spectrometer measurements. We can point this out more clearly in the manuscript.

What is the proportion (e.g. in %) of the total annual DOC and nitrate flux that is exported by the baseflow and by events as defined in the study?

This is roughly visible in figure 10. We will include the exact percentages in the text.

Overall, I agree with the authors opinion that the proposed methodology for separating the baseflow DOC and nitrate from events fluxes is simple and could be used elsewhere. However, the method is in principle based on graphical baseflow separation techniques and is not something new.

Indeed, this technique is not something new. We did not intend to introduce a new technique. We tried to keep it as simple and reproducible as possible.

In the Discussion section, the authors refer to other studies at the same experimental catchment. But there is relatively poor discussion of the results in relation to process understanding. I miss more tangible discussion on how the results of the DOC and nitrate fit into other studies mentioned in the discussion that were done at the same study site. Do they agree well or do the show that some of the explanations proposed in other studies are not in line with the results shown here.

We will improve this section.

Another thing that in my opinion strongly influences preferential flow paths (such as flow paths near the surface or in top soils) is the influence of antecedent wetness, rainfall abundancy and intensity in relation to soli infiltration capacity. Was anything done in this direction? Have authors of this or some other studies in the experimental catchment observed some "boundary conditions" which could be related to the solute concentrations behavior in wet periods and so formation of so called "second peaks"?

We will improve the discussion related to other studies in the catchment.

We have a high porosity and hydraulic conductivity in the catchment – especially in the periglacial slope deposits (Glaser et al. 2016, Jackisch et al. 2016). The second peaks are most likely controlled

by a storage threshold (Martinez-Carreras et. Al 2016). The rainfall intensity is not a controlling factor (Scaini et al. 2016).

Specific comments:

Page 4, lines 16-18: What are the technical characteristics of the UV-Vis spectrometer in terms of the DOC and nitrate concentrations (min, max concentration, detection limits, accuracy, etc.).

We will include that in the revised manuscript.

Page 7, line 12: The rainfall amount of 5 mm seems rather small in order to be considered as a rainfall events. Any additional comment on rainfall losses and rainfall interception, average monthly evapotranspiration from the forested catchment?

A rainfall event of 5mm is already causing a discharge peak in the stream. This is the reason why we chose 5mm as the lower limit for a rainfall event.

We will cite Pfister et al. 2017 for the interception and evapotranspiration

Page 7, lines 9-12: The sentence is unclear and need to be rewritten.

We will rewrite this sentence

Page 7, line 13: I suggest changing: : : :with a minimum 5 h time gap: : :

This is a good suggestion. We will change that.

Figure 5 caption: Does Fig. 5 really show discharge volumes, units are in m3/s?

Fig.5 shows the probability density plots of DOC and nitrate. In Fig. 5 c and d,  the probability density plots are grouped in different discharge classes and the discharge has units of l/s

Page9, line 14: I suggest changing the statement to: : : :similarly increased during first and second peaks.

This is a good suggestion. We will change this sentence.

Page 11, line 5: Authors mention increase of nitrate concentration during second peak. Looking at Fig. 6d, this increase is very small (from approx. 0.8 mg/l (pre-event concentration) to 0.9-1.0 mg/l. I wonder how can this "slight" increase in nitrate concentration be explained in view of UV-Vis spectrometer accuracy?

For some second peaks we took grab samples that were analyzed in the lab. Those results verified the UV-Vis spectrometer measurements.

Page 14, line 2: Is the comment on the results related to Figs. 10 c and d?

Indeed, we need to clarify more specifically which comment is related to which Figure

Page 16, lines 1-2: I believe the discussion on the goodness-of-fit between laboratory DOC concentrations ad in-situ UV-Vis concentrations should be further discussed accosting to my comment provided above.

An offset and a slope different from 1 have to be expected when comparing the in-situ spectrometer measurements with grab samples that are analyzed in the lab (the manufacturer of the spectrometer also points this out in the spectrometer manual). Stream water has a different background matrix compared to calibration standards that are used in the lab. Therefore it is essential, to take grab samples directly in the stream to compare them with the spectrometer measurements. We can point this out clearer in the manuscript.

Page 16, lines 5-7: Please add some references (if available) while mentioning potential problems with the use of measuring equipment in environmental settings different than the presented study site.

We will check for some reference. This sentence is mainly based on oral communication with other researchers that used the same spectrometer and our own experience in two different catchments (those studies are not yet published).

Page 16, line 2: Was the fit between UV-Vis and lab measurement really good (seem my previous comment regarding DOC measurements)?

See our answer to your previous comment. We consider our linear regression as good. We cannot expect a 1:1 line.

Page 16, line 22-23: Are there any field evidences that preferential overland or near surface flow paths really occur at the studied catchment?

Hortonian overland flow has not been observed in the catchment. We have near surface flow from the hillslopes and overland flow from saturated areas. A physically based modeling approach by Glaser et al. 2016 confirmed this. Additional confirmation comes from the reaction of the soil moisture sensors in the catchments (not yet published)

Page 16, line 24: The flushing hypothesis was not originally proposed by Weiler and McDonnell (2006), one of the first that proposed the flushing hypothesis were Hornberger et al. (1994). Therefore I suggest changing the order of the listed references.

This is a good advice. We will change this accordingly.

Page 17, line 9-12: I believe that vice-versa is also true. So the export behavior of DOC or nitrate (or maybe some other dissolved substances) can be very helpful for explaining various runoff components.

Indeed, we fully agree with this comment. Nevertheless, we decided that the focus of the manuscript will be: runoff components explaining export behavior. But we will add here a statement on the fact that vice-versa is also true.

Page 17, Line 34: What is meant by "hot moments"?

We are referring to Krause et al. (2015) who used to expression "hot moments". We introduced this expression on page 3, line 5-7: Catchments generally exhibit a pulsed and highly nonlinear behavior 5

for flow and solute transport. Consequently, monitoring protocols that are too coarse are likely to miss important information during those pulses or so-called hot moments (Krause et al., 2015).

Page 18, lines 15 – 18: Last paragraph of the Conclusion section seems rather general and is in my view not in line with the main theme of the study.

Indeed, this is not a conclusion. It is rather an outlook. We will change that accordingly. We will change the title of section 5 to Conclusions and Outlook.

.

---

## Author Comment (AC3) · 20 Dec 2017

Overview: Recent technological developments have enabled watershed researchers to monitor important biogeochemicals at high frequency for long periods. Sensors that measure both carbon and nitrate are uniquely suited to study the coupling of nitrogen and carbon cycles across varying temporal scales. The authors present a highfrequency, multi-year data set of nitrate and dissolved organic concentration over a range of hydrological and climatological conditions in a forested watershed. The authors leverage this high density dataset to examine rain-fall runoff responses of nitrate and DOC over varying climatological conditions. The authors suggest that antecedent moisture conditions, and as a result, groundwater levels, drives the relative fluxes of nitrate and DOC form the basin.

General comments: This study was well conceived and the results are clearly presented. However, several elements of the manuscript need attention before this study warrants publication.

1. Further, claims that seasonal differences in nitrate/doc fluxes were observed, are not clearly supported by the data presented in this manuscript.

Rather, initial dryness is the direct driver. Consider including data that clearly links the occurrence of preceding dry conditions to season over a longer period of time (longer than two years) to support the seasonal link to hydrologic conditions over the period of this study.

We thank the reviewer for the constructive comments. Indeed, initial dryness is the direct driver for differences in nitrate/doc fluxes through differences in rainfall-runoff processes. In the Weierbach catchment, dryness is generally linked to the growing season. We can proof that with time series of several years of discharge, groundwater and soil moisture. We will either present those time series in the revised manuscript or refer to already published articles about the Weierbach catchment showing these effects.

However, the pattern of dry=growing season, wet=dormant season is not always true. In some years some exceptions existed. We should therefore be more careful when talking about seasonal differences. We will consider that in the revised manuscript.

2. The authors fail to put the implications of the study- that "dry"/"reduced wetness" antecedent conditions results in larger fluxes in nitrate to the stream-into a larger context.

Larger context: We can discuss the implication of climate change to the nitrate export in the Weierbach catchment and in other catchments that behave similar to the Weierbach catchment (catchments with double peak behavior that depend on the initial wetness or catchments where subsurface stormflow is existing but not materialized as a second peak).

3. Consider removing "long-term", as data collected for less than three years hardly justifies the use of this term.

In the context of high-frequency monitoring we consider two years as being long-term measurements. Conventional high-frequency sampling approaches (e.g. with autosamplers) are generally restricted to much shorter time periods of several events. Thus we want to stick to our terminology if the editor supports it.

4. The authors should consider using an outside editing service, given the occurrence of several awkward statements throughout the text (e.g. Pg 2, Line 5; Pg.3 lines 28-30; Pg 12, lines 10-13). We will improve the language quality of the manuscript.

Specific comments: Abstract: Pg 1 Line 11: Define "dry", as this definition is critical in the interpretation of the results as well as applying the findings to other locales and placing the implications of this study into a larger context.

It is difficult to exactly define "dry" as it is a relative term. Dry enough that no second discharge peaks occur. There is an ongoing debate about the dryness threshold for second peaks in the Weierbach catchment. In figure 4 it is possible to compare discharge and the periods with and without second peaks. It might be useful to include groundwater levels and soil moisture content time series.

Introduction: Well cited, however, consider adding Pellerin et al., given the similarity in use of continuous DOC and nitrate sensors to document varying biogechemical yield over different hydrological conditions in a forested watershed.

We will include this article in the introduction.

Methods: I was puzzled as to how the probability density plots were developed, and the source of the data. Please explain in detail which if not all storms were considered and how these distribution plots were generated/modeled.

We will better explain the methods that we used for the probability density plots. There was not modeling involved. We used the entire 2 year time series and divided it into periods with first peaks,

second peaks, baseflow and also into classes of different discharge amounts and derived the probability density plots using R.

The deployment techniques should be more clearly documented to ease duplication of the study. Was the approach modeled after the used in another study? If so please cite. For example, one important aspect is how the sensor cleaned (Birgand et al., 2016, Etheridge et al., 2013)? Pg 5, line 6-7.

We will improve the description of the deployment technique. The sensor was manually cleaned every two weeks with a brush and a detergent that was provided by the manufacturer. Every 3 hours, the sensor was cleaned with air pressure.

This was not modeled after another study.

What was the typical (or max) holding time before analysis of discrete samples?

The samples were filtered and cooled the same day they were taken. We also tried to use autosamplers that were emptied weekly. For nitrate, the samples from the autosamplers were not usable, for DOC it was possible to include the samples.

Results: Consider moving Figure 1 to supplemental material. Characterize the model fits and explain or speculate when/why the outliers tended to occur, especially high residuals for DOC in late 2015.

We guess that the reviewer is referring to figure 2 and not figure 1. We will consider removing figure 2 to supplemental material and try to explain or speculate when the outliers occurred. However, since the other 2 reviewers were quite keen on the results of this figure, we have to balance all suggestions

Placing discrete sample data on the time series Figure 4, would help interpret the limitation of this measurement strategy, i.e. non-linearity due to fouling, light blockage, high turbidity, etc.

There is no pattern visible when displaying the discrete samples in the time series of figure 4. The Weierbach catchment was very suitable for the use of the spectrometer. Due to the low nutrient content, disturbance by biofilm was almost never a problem and the turbidity was generally low.

Please explain the DOC/nitrate data gaps in the summers of 2014/2015.Are these gaps a result of sensor limitations, or were the data otherwise removed? Also, what is the presumed influence of these missing data on cumulative DOC/NO3 export presented in Figure 10.

The gaps in summer 2014 were rather short and were caused by a failure of the sensor due to a cut in power supply. The gaps in summer 2015 were caused by the fact that the stream was nearly dry. No discharge was in the stream for several weeks. This is a natural effect and cannot be considered as missing data but as 0 values.

Figures 3, panel b and Figure 4, panel e and Figure 6, panels d, h, nitrate trace is almost illegible given the color selection. Consider a darker color for the trace and y-axis font.

We will adapt the color.

Figure 4 Justify the presentation of was daily mean values rather than another descriptive statistic (mean,max, etc).

We wanted to show how the variables behave over the 2 years – general behavior like recession periods, periods with high discharge/concentration. Therefore we considered the daily mean as the best way to display the variables in an averaged way.

Pg 9, line3: If the data is sampled to daily mean, how are sub-daily peaks resolved? Pg 11, line 3: replace remarkable to "notable" or equivalent.

Indeed, this is misleading. The subdaily peaks are not visible in figure 4.

Pg.11, Line 3-5/Figure 6. Do the authors speculate on the mechanism for a steady decline/recession in DOC, despite the rise in discharge during second peak?

We do not understand what the reviewer is intending to express.

Pg 12, lines 10-13: be specific about which end members

We will be more specific about that

Consider replacing figure 8, which is unclear and messy and rather plot a few select storms that illustrate hysteresis loops.

We believe that figure 8 contains relevant information. We will try to do it in a less messy way or using a second figure showing individual events

Pg 12, line 18. Change "increasing" to "variable"

Ok.

Pg 13 Lines 10-13.More detail needed here, e.g. what direction were hysteresis loops, for example?

We will provide more details. However, we did not do a detailed analysis of the hysteresis loops. This could be a completely new study.

 Discussion: Pg 16, lines 5-7.Citation suggested here instead of personal experience not included in this study.

ok

Pg 16, line 29-30.Where is the evidence of a rise on groundwater table to support this claim?

We will provide some references.

Pg 17, line 26.Consider changing "concur to the" to "suggests that"

ok

---

## Author Comment (AC4) · 20 Dec 2017

This manuscript reports the use of high frequency measurement of DOC and nitrate using UV VIS spectrophotometer in a forested headwater catchment, Weierbach. The scope of the manuscript is relevant to HESS and presents on the use of relatively new tools –high frequency, automated sensor data to understand the dynamics of DOC and nitrate in the catchment. Rigorous evaluation of these emerging technologies is a pressing need. The aim of this manuscript was to demonstrate that these high frequency automated sensor data add value to understanding event and seasonal dynamics of DOC and nitrate in this catchment. Indeed, the manuscript reaches a confirmatory conclusion that the sensors add value in detecting event peaks and seasonal trends. However, at the end of the manuscript, it is not clear to the reader how trans-ferable these methods are to other catchments; the methods and limitations of the sensors remain poorly articulated in this manuscript.

We thank the reviewer for his constructive comments on this manuscript.

The main objective of the paper was on the export of DOC and nitrate and its relationship with to rainfall-runoff processes. Yet, we acknowledge that more information about the limitations of the sensor is beneficial. We will do that in the revised manuscript in more detail.

In particular, a number of concerns are raised concerning detail of the methods and assumptions. These concerns include 1) details on methods for filtering samples and calibration of sensor seemed inadequate to reproduce,

We will improve that in our manuscript.

2) the adequacy of use of linear regression for calibrating sensor data to manual grab samples

We followed the instructions of the manufacturer of the spectrometer that advices a linear calibration. The results shown in Figure 2 support our decision to use a linear regression.

and well as 3) use of non-conservative tracers as end members to attribute to sourcing.

We are aware of the limitations of DOC and nitrate as tracers – therefore we do not use them for classical mixing calculations. The information that we use for the sources does not only come from DOC and nitrate, but is also based on previous work in the catchment by other authors. We will discuss the available Weierbach literature on this topic in more detail. However, the main goal of this manuscript is on export of DOC and nitrate and not on sources.

Most studies in the literature use concentrations obtained from grab samples as the known concentration values (X axis) and sensor data as the response variable (Y axis). Uncertainty in both the x and y as well as self-correlations (see Worrall et al. 2015 should be given consideration and

other models (possibly orthogonal regression or other models (see Vaughan et al. 2017)) should be adopted to address uncertainty in x and y values.

We could change the x- and y values, but this will not considerably alter the results on the correlation between the two variables. Self-correlations should not be an issue, as the water samples were taken at bi-weekly interval. Since we used the regression to test how well the spectrolyser can describe lab samples, we do not see the necessity to consider the uncertainty of the measurements.

Finally, this manuscript is not as fluent as it could be, and it could be more concise. There were numerous grammatical and tense agreement issues as well as awkward statements that made accessing the results and discussion a bit challenging. I believe that there could be better use of figures and some figures could be combined and/or possibly put into supplemental materials.

We will improve this in the revised version.

Standard hydrograph separation techniques should be used or better referenced

We will improve the reference.

Adequate referencing of description of prior research in the Weierbach is needed in the introduction to understand the research gap but also the rationale behind single and double peak separation. Some of these studies are cited in the discussion but this discussion comes too late. Finally, putting these data into context of other studies is desperately needed to improve the manuscript.

We will improve that in the manuscript. However, previous research is somewhat ambiguous about the underlying processes generating the first and second peaks are.

Specific comments

Page 4, line 1-5: Hypotheses are a bit awkward; what is meant by individual concentration signals?

We hypothesis, that the response of DOC and nitrate concentrations during rainfall-runoff events help us to identify the relevant flow paths in the catchment.

Page 5, line 4: Need more details on global calibration provided by manufacturer.

The manufacturer does not provide more details on the global calibration. However, we will better explain what is meant by the term global calibration.

Page 5, line 7: filter through what kind of filter? Need more details here. And were the samples run for NPOC or TC-TIC? Then you are sparging the samples?

We will provide more details on that.

Page 5, line 10: if only using nitrate samples, how did you cover the full range of nitrate concentrations at high discharges?

The maximum concentration that we collected with the grab samples was 1.2 mg/l. This covered the vast majority of the measured nitrate concentration. Indeed, we did not cover all the whole concentration range with our grab samples. In some rare cases, the measured nitrate concentration reached almost 2 mg/l.

Page 5, lines 10-15: Most studies found in the literature (e.g. Vaughan et al. 2017) use concentrations obtained from grab samples as the known concentration values (X axis) and sensor data as the response variable (Y axis). This is the reverse of what has been done in Figure 2. Uncertainty in both the x and y as well as selfcorrelations (see Worrall et al. 2015 should be given consideration and other models (possibly orthogonal regression or other models (see Vaughan et al. 2017)) should be adopted to address uncertainty in x and y values. Were the residuals of the line regression examined to determine if they are normally distributed? They appear at bit heteroscedastic, especially for DOC.

see comment and response above

Finally, this section is awkward and could use a rewrite. Here the manuscript starts to switch back and forth. For example, it should read DOC concentrations of grab samples "were" linearly correlated to stay consistent with past tense used throughout paragraph.

We will improve the language in the revised manuscript.

Page7, line 3: More specifics on rating curve are needed. Are these published rating curves or derived from other publications. If so, please reference or provide more details.

This is a longterm gauging station operated by LIST. The rating curve is based on continuous water pressure (water level) measurements and salt dilution discharge measurements. If we had used discharge data from national gauging sites, we could also not provide the required details as they are not available in most cases.

Page 7, line 4-19: This section needs more referencing to prior research in the catchment as well as referencing for hydrograph separation. If these rainfall runoff characteristics are not reported elsewhere (but it seems like they are based on discussion), these are results and should go into the results section. Also, referencing to standard hydrograph separation or using accepted hydrograph separation would add utility to this paper.

We will explain that in a better way in the revised manuscript.

Page 7, line 24: Figure 3 could be perhaps moved to supplemental if adequate referencing is use.

We believe that figure 3 is of importance to explain the methods.

Page 12, line 4: here the mansuscript using DOC and nitrate endmembers and infers sources without providing additional conservative tracers to determine sources. Did you collect or analyze samples for other conservative tracers for both hydrograph separation and end member mixing analysis. Using a conservative tracer is a central assumption of these analyses (see Barthold et al. (2011)) such that the validity of interpretation of sourcing in this manuscript is questions without additional evidence supporting this conclusion.

Although we are not completely sure about the sources, we do not agree that we necessarily need end-member mixing analysis to determine the sources. We are referring to Glaser et al. (2016), that identified sources based on a modeling approach.

Page 12, line 16. Suggest change paragraph to past tense.

ok

Page 13, line 10, relevant? Suggest a different word choice, suggest substantial

ok

Page 16, line 5: this line refers to other studies and limitation of the sensors. Can you reference to these studies? Perhaps this would be a better method paper if these data were also reported to show the limitation of the sensors?

We will improve the references. However, some of the knowledge about the limitation of the sensor is based on oral communication with other research groups that have use the same sensor. Besides, we do not consider the manuscript as being a method paper.

Page 16, line 8-19: This seems slightly redundant to methods. Perhaps include this section or shortened version thereof in methods. Can you reference other studies that have done this?

We believe that the methods should be discussed in the discussion section. We will improve the referencing.

Page 16, line 20-34. Again, tenses are switched in the paragraph. I'd suggest keeping in past tense.

We will improve that.

Page 17, line 4, suggest delete "have"

ok

Page 17, line 5: awkward sentenceˇAˇ Tsuggest revise

ok

Page 17, line 12 and starting line 20. Here you refer to other studies that have been conducted in the catchment identifying the single and double peaks. This is a strong

rationale for your study, approach, and analysis. I think some of this needs to come up to the introduction or methods.

We will try to find a better balance between introduction, methods and discussion.

Page 17, line 32: Suggest delete second sentence starting "By only: : :" repeat of same sentence.

ok

Figures: Figure 8 and 9 could be combined into one figure with different panels.

We will have a look if that is possible and improve the figures.

Technical corrections: Page 4, Line 8: spell out masl the first time

ok

Page 4, line 9 and 10: italicize Fagus sulvatica and Picea abies

ok

Page 4, Line 10: delete period (.) before (Glaser et al., 2016)

ok

Page 4, line 13: Suggest change "The" to "A" and change "is causing: to "results in"

ok

Page 6, line 3, suggest delete "for" before throughfall

ok

Page 6, line 7, suggest add "(Figure 1)" after location

ok

Page 9, line 6: delete "the" before nitrate and change to "had" to "were" and delete "levels".

ok

Page 9, line 7: Suggest sodify sentence to read "Nitrate concentrations decreased during recession periods, as observed in spring 2014 and 2015: : :

ok

Page 11, line 5: suggest change "reaction" to "response" because you can not infer reaction.

ok

Page 16, line 2: delete "have"

ok

References referred to within comments:

Vaughan, M. C. H., et al. (2017), High-frequency dissolved organic carbon and nitrate measurements reveal differences in storm hysteresis and loading in relation to land cover and seasonality, Water Resour. Res., 53, 5345–5363,

doi:10.1002/2017WR020491.

Worrall, F., Burt, T. P. & Howden, N. J. K. The problem of self-correlation in fluvial flux data – The case of nitrate flux from UK rivers. Journal of Hydrology 530, 328–335 (2015).

Barthold, F. K., C. Tyralla, K. Schneider, K. B. Vaché, H.-G. Frede, and L. Breuer (2011), How many tracers do we need for end member mixing analysis (EMMA)? A sensitivity analysis, Water Resour. Res., 47, W08519, doi:10.1029/2011WR010604.